# Standardizing Distress Analysis: Emotion-Driven Distress Identification and Cause Extraction (DICE) in Multimodal Online Posts

**Gopendra Vikram Singh**[*]**, Soumitra Ghosh**[*]**, Atul Verma, Chetna Painkra**
and **Asif Ekbal**

Department of Computer Science and Engineering, IIT Patna, India
{gopendra_1921cs15,asif}@iitp.ac.in,
{ghosh.soumitra2,atul.verma.a3,chetnapaikra55}@gmail.com

## Abstract

Due to its growing impact on public opinion, hate speech on social media has garnered increased attention. While automated methods for identifying hate speech have been presented in the past, they have mostly been limited to analyzing textual content. The interpretability of such models has received very little attention, despite the social and legal consequences of erroneous predictions. In this work, we present a novel problem of *Distress Identification and Cause Extraction (DICE)* from multimodal online posts. We develop a multi-task deep framework for the simultaneous detection of distress content and identify connected causal phrases from the text using emotional information. The emotional information is incorporated into the training process using a zero-shot strategy, and a novel mechanism is devised to fuse the features from the multimodal inputs. Furthermore, we introduce the first-of-its-kind *Distress and Cause annotated Multimodal (DCaM)* dataset of 20,764 social media posts. We thoroughly evaluate our proposed method by comparing it to several existing benchmarks. Empirical assessment and comprehensive qualitative analysis demonstrate that our proposed method works well on distress detection and cause extraction tasks, improving F1 and ROS scores by 1.95% and 3%, respectively, relative to the best-performing baseline. The code and the dataset can be accessed from the following link: https://www.iitp.ac.in/~ai-nlp-ml/resources.html#DICE.

## 1 Introduction

The exponential expansion of microblogging sites and social media not only empowers free expression and individual voices, but also allows individuals to exhibit anti-social conduct (ElSherief et al., 2018), such as cyberbullying, online rumours, and spreading hate remarks (Ribeiro et al., 2018). Abusive speech based on race, religion, and sexual orientation is becoming more common (Karim et al., 2020). Automatic identification of hate speech and raising public awareness are critical tasks (Karim et al., 2020). Manually evaluating and validating a large volume of web information, on the other hand, is time-consuming and labor-intensive.

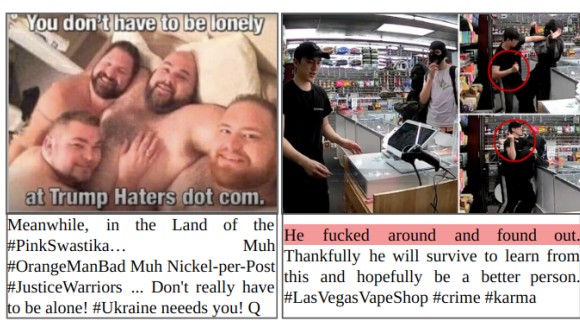

Figure 1: Sample *Distressed* posts from our *DCaM* dataset. Span annotation is highlighted in red.

Modern language models excel over traditional machine learning and neural network-based approaches but lack transparency in output transformation, posing limitations in domains, such as the military, medical research, and internet content monitoring. Robust models for monitoring distressed content online require multimodal inputs. In our "DCaM" dataset, Figure 1 highlights the significance of multimodality and span annotations in comprehending distress content. While both posts are labeled as "distressed," the first post may not offer sufficient information based on textual content alone. However, the second post, with both picture and text, provides clarity, and the span annotation aids in analyzing the manifestation of distress.

This necessitates a shift in viewpoint away from performance-based models and toward interpretable models. We address model explainability by jointly learning the target classification of a multimodal social media post as *Distressed* or *Non-*

---

[*] These authors contributed equally to this work and are the joint first authors.

*distressed* and extracting the reasons for the classification decision (for the *Distressed* class) from the textual input. The prime focus of this study is to comprehend the causes associated with any form of offensive content (hate, offensive, abusive, etc.). We club all the connotations of offensive content under the category *distressed*.

The main contributions are summarized below:

1. We propose the novel task of Unified *Distress Identification and Cause Extraction (DICE)* from multimodal online posts.

2. We develop a multi-task deep framework for the simultaneous detection of distress content and identify connected causal phrases from the text using emotional information.

3. We devise a zero-shot strategy to dynamically incorporate emotional information into training and propose a novel fusion mechanism to infuse the features of multimodal inputs.

4. The first *Distress and Cause annotated Multimodal (DCaM)* corpus is created consisting over 20,764 social media posts.

5. Resources are open-sourced to aid research.

The rest of the paper is organized as follows. Section 2 summarises some previous works in this area. We discuss the dataset preparation in Section 3. Section 4 addresses our proposed methodology in depth, followed by the results and analysis in Section 5. Finally, we conclude our discussion in Section 6 and define the scope of future work.

## 2 Related Work

Several approaches have been suggested to identify online hate speech (Burnap and Williams, 2016; Zhang et al., 2018; Qian et al., 2018). The current interest in hate speech research has led to the availability of datasets in several languages (Sanguinetti et al., 2018; Ousidhoum et al., 2019) and different computational ways to counteract online hate (Mathew et al., 2019; Aluru et al., 2020). Text-, user-, and network-based traits and characteristics that identify bullies have been extracted in (Chatzakou et al., 2017). Deep learning Lundberg and Lee (2017); Founta et al. (2019) has been used extensively to identify hate speech keyword identification, sexism, bullying, trolling, and racism.

Recent research on identifying hate speech has made use of deep learning techniques, including neural networks (Han and Eisenstein, 2019) and

word embedding techniques (McKeown and McGregor, 2018). Recent models based on Transformers (Vaswani et al., 2017) have had extraordinary success. Since this is essentially a classification problem, BERT (Bidirectional Encoder Representations from Transformers) (Devlin et al., 2018) has found widespread use in the field of hate speech identification. Ranasinghe et al. (2019) showed that a BERT-based model performed better than models based on recurrent neural networks (RNNs). Zaidan et al. (2007) first proposed the use of rationales, where human annotators highlight text that supports their classification decision. This work was enhanced by Yessenalina et al. (2010) to provide self-generating rationales. An encoder-generator system for quality rationales without annotations was presented in Lei et al. (2016). Mathew et al. (2021) used dataset rationales to fine-tune BERT to address bias and explainability.

Recent research has shifted towards accommodating multimodal content, with a focus on detecting hate speech and objectionable material in various media. Gandhi et al. (2019) developed a computer vision-based technique for identifying offensive and non-offensive images in large datasets. Kiela et al. (2020) introduced a novel challenge for multimodal hate speech detection in Facebook memes. Rana and Jha (2022) employed the Hate Speech Recognition Video Dataset to identify emotion-based hate speech in a multimodal context. Karim et al. (2022) presented a dataset for detecting hate speech in Bengali memes and text. Fersini et al. (2022) discussed SemEval-2022 Task 5, focusing on identifying misogynous memes through text and images, including sub-tasks for recognizing misogynous content and categorizing types of misogyny. Hee et al. (2022) investigated multimodal hateful meme detection models and their ability to capture derogatory references in both images and text. Additionally, Cao et al. (2022) introduced PromptHate, a model that leverages pretrained language models with specific prompts and examples for hateful meme classification.

Even though multimodal studies on offensive content have gotten a lot of attention, this study is the first to look at how to find distressed content on social media and figure out what caused it. Additionally, this work presents the first *Distress and Cause annotated Multimodal (DCaM)* corpus of social media posts to the research community.

| Datasets | Labels | Total Size | Language | Multimodal? | Rationales? |
|---|---|---|---|---|---|
| Waseem and Hovy (2016) | Racist, Sexist, Normal | 16,914 | English | x | x |
| Davidson et al. (2017) | Hate Speech, Offensive, Normal | 24,802 | English | x | x |
| Founta et al. (2018) | Abusive, Hateful, Normal, Spam | 80,000 | English | x | x |
| Ousidhoum et al. (2019) | Labels for five different aspects | 13,000 | English, French, Arabic | x | x |
| Mathew et al. (2021) | Hate Speech, Offensive, Normal | 20,148 | English | x | ✓ |
| *DCaM* (ours) | Distressed (Hate-Offensive-Abusive), Non-distressed | 20,764 | English | ✓ | ✓ (causes) |

Table 1: Comparisons of different distress datasets.

# 3 Dataset

We discuss the data collection and annotation details in the following subsections.

## 3.1 Data Collection

We collect our dataset from sources where previous studies (Davidson et al., 2017; Zannettou et al., 2018; Mathew et al., 2021) on hate speech have been conducted: Twitter and Gab[1]. The data was scraped from the top 5 trending topics on Twitter using selenium[2] to reduce the effects of sample bias. As for Twitter, we selected the top 10 percent of all collected tweets between October 2022 and December 2022. Using the textual mode of scraped tweets, we generated a list of the most frequent words, which we then used as tags to gather the posts from Gab. Please refer to Appendix Section A.1 for details on data collection from Gab, including keywords used for the *DCaM* dataset (see Table 8). To compile this data, we scoured Gab for posts between November and December 2022. Posts that have been deleted and reposted are not considered. We also remove links from posts to ensure that annotators can access all relevant information. A number of distress datasets are compared in Table 1.

## 3.2 Data Annotation

To ensure the dataset consists of only English posts, we used the TextBlob library for language detection and included only those identified as English. Additionally, non-English posts were flagged and excluded during annotation. Annotators were informed about the presence of hate or offensive content beforehand. Annotation guidelines[3] from Poria et al. (2021); Ghosh et al. (2022c) were provided to assist annotators in understanding the classification and span annotation tasks. Each post was annotated

---

[1] https://twitter.com/, https://gab.com/
[2] https://pypi.org/project/selenium/
[3] The annotation guidelines are discussed in Section A.2 of the Appendix

by five annotators[4] (*DI* task), and then majority voting was applied to decide the final label.

There are two kinds of annotations in our dataset. First, whether the post is *Distressed* or *Non-distressed* post. Second, if the text is considered as *Distressed* by majority of the annotators, we ask the annotators to highlight parts of the text that include terms that might be a plausible basis for the provided annotation. These span annotations help us to delve further into the manifestations of hatred or offensive speech.

| | **Twitter** | **Gab** | **Total** |
|---|---|---|---|
| *Distressed* | 3248 | 5210 | 8458 |
| *Non-distressed* | 7066 | 5240 | 12306 |
| **Total** | 10314 | 10450 | 20764 |

Table 2: Dataset details

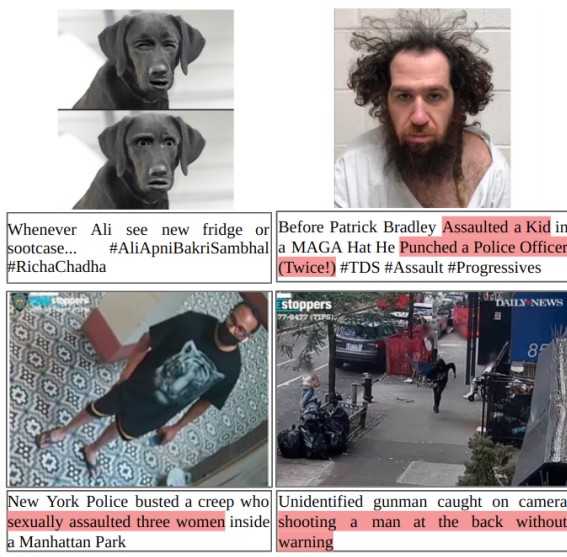

Figure 2: Samples from our dataset

For the *Distressed Identification* task, the Krippendorff's $\alpha$ for the inter-annotator agreement is 0.66 which is much higher than other hate speech

---

[4] 2 Ph.D. Linguistics degree holders, 2 Ph.D. students, and 1 Undergraduate student from the Computer Science discipline

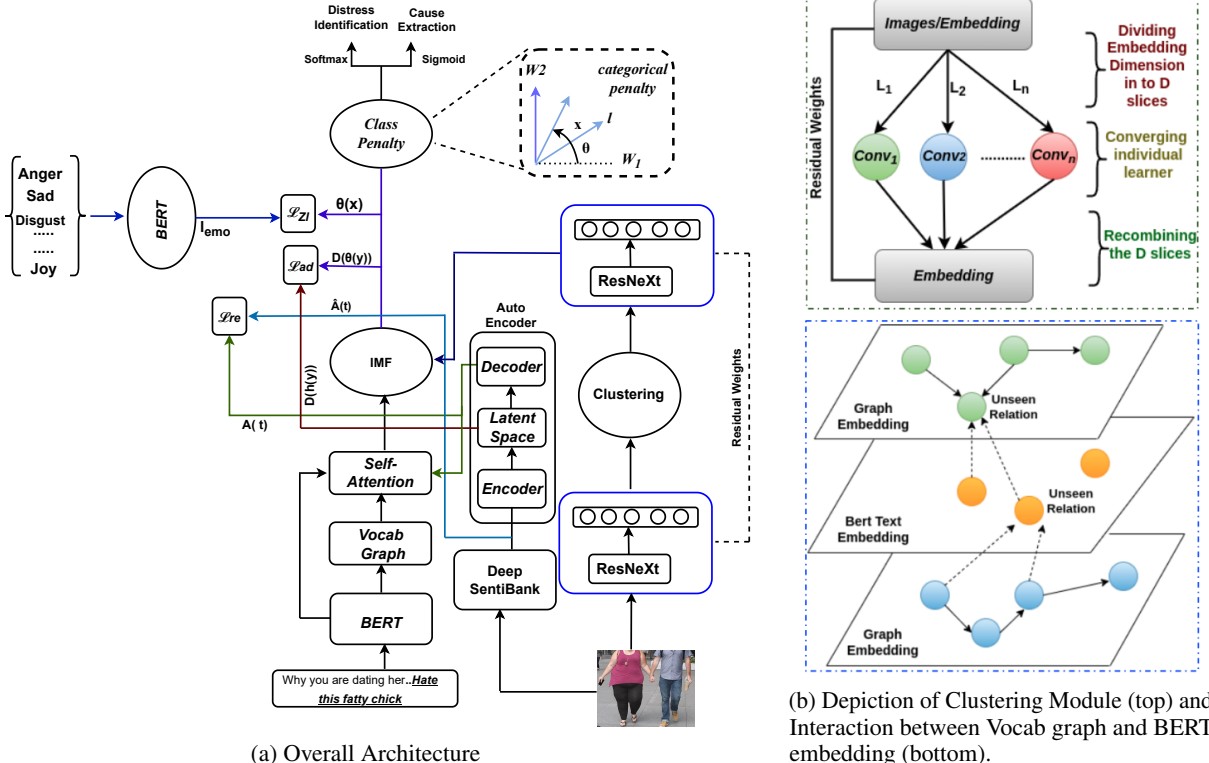

(a) Overall Architecture

(b) Depiction of Clustering Module (top) and Interaction between Vocab graph and BERT embedding (bottom).

Figure 3: Illustration of the proposed *Depression Identification and Cause Extraction (DICE)* framework

datasets (Ousidhoum et al., 2019; Mathew et al., 2021). Following the work in (Poria et al., 2021; Ghosh et al., 2022c), we marked at most 3 causal spans for a *distressed* post in the dataset. The final causal span is marked using the span-level aggregation approach detailed in (Gui et al., 2016). We use the macro-F1 measure to assess inter-rater agreement based on previous work on span extraction (Poria et al., 2021; Ghosh et al., 2022c), and achieve an F1-score of 0.73, suggesting that the annotations are of high quality. Table 2 contains further information about the dataset obtained. Figure 2 shows samples of our dataset. The average number of tokens highlighted per distressed post is 8.55, and the average token per post is 25.43.

## 4 Methodology

In this section, we illustrate our proposed *DICE* framework, which is a multitask system for *Depression Identification and Cause Extraction* from multimodal social media posts. The system employs a zero-shot strategy to dynamically incorporate emotional information into training and presents a novel fusion mechanism to infuse the features from the multimodal inputs. The overall architecture of the proposed method is shown in Figure 3a.

### 4.1 Problem Formulation

Given a post P = $[s_1, \cdots s_i \cdots, s_p]$ composed of a sequence of sentences (s), and each utterance can be further decomposed into a sequence of words. *p* indicates the number of sentences in the post. The objective is to determine if the post is *distressed* or not (0 or 1) and to extract every plausible causal span that supports the prediction.

### 4.2 Proposed *DICE* Framework

**Textual Encoder.** Our textual encoder uses BERT followed by an ontology-based word graph. BERT extracts local information from a text. Ontology is the backbone of knowledge graphs (KGs) (Song et al., 2022), which give meta-data descriptions to guide the creation and completion of knowledge graphs. Additionally, relation descriptions contain semantic information that can be used to represent relations. During Graph Neural Network (GNN) message transmission, we embed text within ontology nodes. First, all the nodes are embedded using node embedding and text embedding as follows:

$$h_o = h_o \mathcal{W}_o^E \quad and \quad h_t = \sum_{n=1}^{N} x_i \mathcal{W}_t^E \quad (1)$$

where $\mathcal{W}_t^E$ is word text embedding (BERT), $\mathcal{W}_o^E$ is graph embedding, $x_i$ depicts a node (representing a word). $h_o$ is a concept in ontology. Figure 3b illustrates the interaction between the vocab graph and BERT embedding to establish relationships. Our method enriches the text-embedding and graph-embedding space, enabling the identification of previously unseen relationships between graph embeddings of the head and tail.

$$r_a = \sum_{n=1}^{N} g(h) \tag{2}$$

where, $r_a$ is aggregate relationship, g(*) is aggregate function, and N is neighboring nodes for the missing node.

**Image Encoder.** We use ResNet[5] to capture facial expressions and visual surroundings for rich emotional indicators from the image in the input post. We separated the embedding dimensions and image data into groups to simplify the problem and make better use of the complete embedding space. Each learner will create a unique distance metric using just a subspace of the original embedding space and a portion of the training data. By segmenting the network's embedding layer into $D$ consecutive slices, we are able to isolate $D$ unique learners inside the embedding space. After learner solutions converge, we aggregate them to obtain the whole embedding space. The merging is accomplished by recombining the slices of the embedding layer that correspond to the D learners. To ensure uniformity in the embeddings produced by various learners, we then perform fine-grained tuning across the entire dataset. The merged embeddings may be hampered by the gradients, which resemble white noise and would hinder training performance. This is called the "shattered gradients problem". To address this, *residual weights* (Balduzzi et al., 2017) provide the gradients with some spatial structure, which aids in training, as shown in Figure 3b.

**Inter-modal Fusion (IMF).** The *IMF* module exchanges information and aligns entities across modalities (text and image) to learn joint inter-modality representations. Figure 4 illustrates the mechanism of inter-modal fusion.

*Text infused visual features (and vice-versa).* We use an external word embedding model to build high-level representations ($\mathcal{T}_i^{'}$) for an image-text

[5] https://github.com/josharnoldjosh/ResNet-Extract-Image-Feature-Pytorch-Python

pair consisting of $\mathcal{I}_i$ and $\mathcal{T}_i$. Cross attention is employed to combine the textual and visual features to create the *Text infused visual features* ($T_V$). Taking into account the spatial properties of the channel-wise features, the query vectors (Q) are generated by convolution with N*kernels on each channel of $\mathcal{I}_i$ and then averaging (avg pooling) the feature maps as illustrated in Figure 4. Similarly, we construct the *Visual infused textual features* ($V_T$) by exchanging $\mathcal{I}_i$ and $\mathcal{T}_i$. In particular, the key vectors (K) are produced by convolution with N*kernels on each channel of $\mathcal{I}_i^{'}$ and then averaging (average pooling) the feature maps.

*Cross-Attention.* First, we take the query vector from one modality (say image, I) and the key/value pair from the other (say text, T). To examine how text affects the image vector, we feed the query ($I_q$) and textual key/value to self-attention ($selfAtt(.)$).

$$I_q = Query(I)$$
$$T_k, T_v = Key(T), Value(T) \tag{3}$$
$$S_{TA} = selfAtt(T_k, T_v, I_q)$$

We filter noise from the output of the self-attention using the forget gate ($\sigma$) and concatenate it with the linear layer's residual (c.f. Figure 4).

$$G_{TI} = Concat(linear(S_{TI}), \sigma(linear(S_{TI}))) \tag{4}$$

Finally, we pass the representations of all the modalities (i.e., text, and image) through another self-attention to know how much the image vector will be impacted by text [$Cross_{TI} = SA(G_{TI}, I_q)$] Please note that bolded **I** in $Cross_{TI}$ represents the impacted modality (i.e., I). Similarly, we compute $Cross_{IT}$ and concatenate all of them to obtain the cross-attentive multimodal features.

*Final Fusion.* Although, the $T_V$ and $V_T$ can independently conduct image-text multimodal recognition, to further enhance the model's performance, we apply self-attention to fuse the two aforementioned feature vectors.

**Class Penalty.** The inter-modal fusion unit receives a class penalty value to help the model understand the link between a unified distress label and the input post. This improves the prediction of start and end tokens. The equations below origi-

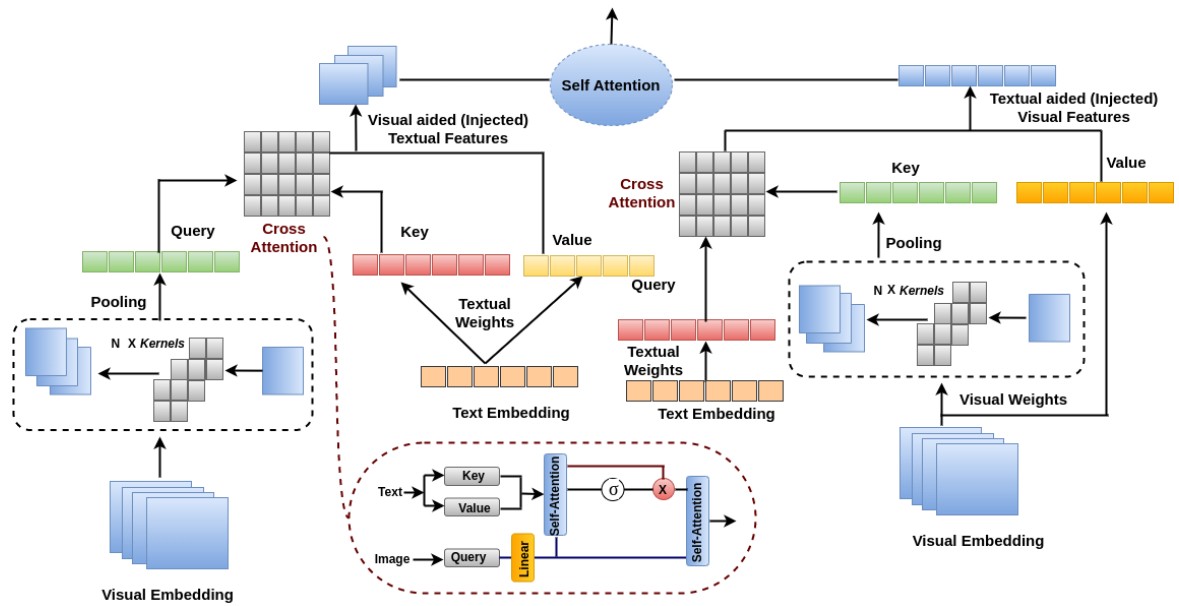

Figure 4: Inter-modal Fusion

nally represent softmax and sigmoid:

$$\mathcal{L} = -\frac{1}{b_s} \sum_{i=1}^{b_s} \log \frac{\exp^{\mathcal{W}l_i + b_i}}{\sum_{j=1}^{N} \exp^{\mathcal{W}l_j + b_j}} \quad (5)$$

$$\mathcal{L} = -\frac{1}{b_s} \sum_{i=1}^{b_s} \frac{1}{\exp^{\mathcal{W}l_i + b_i}} \quad (6)$$

Where, $l_i \in \mathbb{R}^d$ is the feature of $i^{th}$ sample; $b_s$ is batch size; $b_i$ and $b_j$ denote the bias; and $\mathcal{W} \in \mathbb{R}^{d*n}$ denotes the weight matrix. Information extraction tasks are notoriously difficult to find the decision boundary for the start and end markers of a span, and a basic softmax/sigmoid classifier cannot manage this distinction. Some samples may be misclassified due to the classification boundary's ambiguity. This may require a faster convergence rate. We use the Insightface loss technique (Deng et al., 2019) to normalize the feature $l_i$ and weight matrices. $\mathcal{W}$ to assess feature similarity based on the angle difference by which it maps the vector more closely. To converge the feature, it adds a penalty value $x$ to the angle.

$$\mathcal{L}_{u1} = -\frac{1}{b_s} \sum_{i=1}^{b_s} \log \frac{\exp^{a(cos(\theta+x))}}{\exp^{a(cos(\theta+x))} + \sum_{j=1}^{N} \exp^{a(cos(\theta))}} \quad (7)$$

$$\mathcal{L}_{u2} = -\frac{1}{b_s} \sum_{i=1}^{b_s} \frac{1}{\exp^{a(cos(\theta+x))} + \exp^{a(cos(\theta))}} \quad (8)$$

where $\mathcal{L}_{u1}$ and $\mathcal{L}_{u2}$ is updated loss functions for softmax and sigmoid, respectively, $\theta$ denotes the angle between weight $\mathcal{W}$ and feature $l$ and a denotes the amplifier function.

**Emotion Features.** We consider Ekman's (Ekman, 1992) emotion classes and initialize them with the BERT (Devlin et al., 2018) vectors to represent their semantic features.

*Reconstruction Loss.* An auto-encoder reconstructs adjective-noun pair (ANP) features[6] and produces latent features while maintaining emotion information in the learned latent space to match label and ANP feature structures. By optimizing the following loss function, the auto-encoder input $(A)$ and output $(\hat{A})$ must be sufficiently close to identify its parameters.

$$\mathcal{L}_{re} = ||\hat{A}(IMF(a,t)) - A(IMF(a,t))||_2^2$$

Also, optimizing this loss results in lower-dimensional input features and high-accuracy feature reconstruction.

*Adversarial loss.* Our objective is to maintain the discriminative capacity of the combined fea-

---

[6]To begin, we employ mid-level semantic representations of ANP features for the creation of an intermediary latent space. When provided with a training image, we opt for the application of the pre-trained ANP detector, DeepSentiBank (Chen et al., 2014) , to extract the ANP feature . To establish a proficient latent space conducive to a concise representation of the original affective features , we embrace the utilization of an auto-encoder model.

| Modality | Distress Identification | | Cause Extraction | | | | |
|---|---|---|---|---|---|---|---|
| | F1 (%) | ACC. (%) | FM | PM | HD | JF | ROS |
| DICE (T) | 86.12 | 86.54 | 38.74 | 40.51 | 0.66 | 0.82 | 0.84 |
| DICE (I) | 70.15 | 72.11 | 28.41 | 31.28 | 0.52 | 0.71 | 0.71 |
| ViL-BERT CC | 83.75 | 84.78 | - | - | - | - | - |
| Visual BERT COCO | 85.27 | 86.30 | - | - | - | - | - |
| DICE (T+I) | **87.71** | **89.53** | **41.31** | **45.48** | **0.69** | **0.85** | **0.88** |

(a) Results across different modalities. Here, T: Text, I: Image

| Models | KC | Inf | F |
|---|---|---|---|
| BERT-HateXplain | 2.98 | 2.73 | 3.12 |
| SpanBERT | 3.26 | 3.47 | 3.55 |
| CMSEKI | 3.41 | 3.43 | 3.63 |
| *DICE (Proposed)* | **3.69** | **4.07** | **3.98** |

(b) Results of human evaluation. Here, KC: Knowledge Consistency, Inf: Informativeness, F: Fluency

Table 3: Results of the *DICE* framework on the *DCaM* dataset. Values in bold are the maximum scores attained.

ture of the text and visual i.e.$A(IMF(a, t))$, and combine it with the rich emotional structural data contained in feature $\phi(l_{emo})$. This is accomplished by using an adversarial restriction that seeks to trick a discriminator network $\mathcal{D}$ such that the output $A(IMF(a, t))$ features are as comparable as the ANP features:

$$\mathcal{L}_{adv} = \mathcal{E}_y(\log \mathcal{D}(h(y))) - \mathcal{E}_y(\log \mathcal{D}(\theta(y)))$$

Where $\theta(y)$ defines the combined feature of text and image i.e. MF (a,t); and h(y) defines the latent feature space. The generator network (auto-encoder) minimizes this loss to learn how to generate emotionally rich labels that closely match the ANP features, ensuring accurate label generation.

*Zero-shot loss*. Suppose $\theta(x)$ defines the combined feature of text and image i.e. MF(a,t), and $\phi l_{emo})$ defines the semantic feature of the label. The objective here is to reduce the distance between these two using the following function:

$$\mathcal{L}_{zl} = ||\theta(MF(x) - \phi l_{emo})||_2^2$$

The zero-shot loss enhances the generation of accurate and emotionally rich labels by aligning the combined feature of text and image with the semantic feature of the emotion classes.

*Joint Loss*. The model is trained using a unified loss function defined below:

$$\mathcal{L}_{joint} = \mathcal{L}_{adv} + \mathcal{L}_{zl} + \mathcal{L}_{re}$$

*Emotion Label Prediction*. For a given post (text+image), our model will classify the labels using a simple nearest neighbour (NN) search. Let us suppose that the post and labels are fed into the embeddings to obtain $\theta(MF(a, t))$ and $\phi(l_{emo})$.

$$||\theta(MF(a, t)) - \phi(l_{emo})||_2^2$$

### 4.2.1 Calculation of Final Loss

As illustrated in equation 9, the model is trained using a unified loss function. For both the *DI* and

*CE* tasks, we employ binary cross-entropy loss.

$$L = \sum_\omega W_\omega L_\omega \quad (9)$$

Here, $\omega$ represents the two tasks, *DI* and *CE*. The weights ($W_\omega$) are updated using back-propagation for specific losses for each task.

## 5 Experiments and Results

This section discusses the results and the analysis. Due to space constraints, we discuss the experimental setup in Section A.3 and the evaluation metrics in Section A.5.1 in the Appendix.

### 5.1 Baselines

Our framework combines distress identification and cause extraction into a single automated system, utilizing classification and span detection. Due to the lack of suitable multimodal baselines with similar objectives, existing automated systems were used for evaluation. We compare our proposed *DICE* approach and the presented *DCaM* dataset against various baselines, including BiRNN-Attn (Liu and Lane, 2016), CNN-GRU (Zhang et al., 2018), BiRNN-HateXplain (Mathew et al., 2021), BERT (Liu et al., 2019a), BERT-HateXplain (Mathew et al., 2021), SpanBERT (Liu et al., 2019b), and CMSEKI (Ghosh et al., 2022b). To thoroughly evaluate our approach on multimodal inputs, we employed two widely-used multimodal baselines, ViLBERT CC (Lu et al., 2019) and Visual BERT COCO (Li et al., 2019), to assess the distress identification task in our dataset. We discuss the baselines briefly in Section A.4 of the Appendix.

### 5.2 Results and Analysis

Table 3 shows the results of the proposed *DICE* framework on the introduced *DCaM* dataset. Specifically, we show the modality-varying results

| Models | Distress Identification (DI) | | Cause Extraction (CE) | | | | |
|---|---|---|---|---|---|---|---|
| | F1 (%) | ACC. (%) | FM | PM | HD | JF | ROS |
| BiRNN-Attn (Liu and Lane, 2016) | 75.47 | 76.71 | 26.21 | 30.41 | 0.51 | 0.69 | 0.74 |
| CNN-GRU (Zhang et al., 2018) | 76.13 | 78.11 | 27.13 | 30.92 | 0.53 | 0.70 | 0.75 |
| BERT (Liu et al., 2019a) | 81.78 | 82.41 | 31.32 | 36.38 | 0.58 | 0.75 | 0.78 |
| BiRNN-HateXplain (Mathew et al., 2021) | 77.71 | 78.58 | 28.51 | 32.16 | 0.54 | 0.72 | 0.76 |
| BERT-HateXplain (Mathew et al., 2021) | 82.69 | 83.19 | 33.41 | 37.73 | 0.60 | 0.77 | 0.79 |
| SpanBERT (Joshi et al., 2020) | 82.83 | 83.66 | 33.98 | 38.22 | 0.61 | 0.77 | 0.79 |
| CMSEKI (Ghosh et al., 2022a) | 84.17 | 85.31 | 36.39 | 38.22 | 0.64 | 0.80 | 0.81 |
| *DICE (Proposed)* | **86.12** | **86.54** | **38.74** | **40.51** | **0.66** | **0.82** | **0.84** |

Table 4: Results from the *DICE* model and the various baselines. Here, the bolded values indicate maximum scores.

in Table 3a. The bi-modal (Text+Image) configuration yields the best results, followed by the uni-modal network. The textual modality outperforms the others when compared independently, as texts have less background noise than visual sources. For the similar tasks, our results are consistent with prior studies (Hazarika et al., 2018).

**Human evaluation.** A qualitative human review was conducted on 300 randomly selected posts from the test dataset to assess the model's identified causes. The assessment used three well-defined measurements (Singh et al., 2022), with scores ranging from 0 to 5 based on Fluency, Knowledge Consistency, and Informativeness[7]. Scores of 0 were given to the most incorrect responses, while the best responses received a score of 5. In Table 3b, it can be seen that, compared to the various baselines, the proposed framework has done well for all the manual evaluation measures. Our suggested approach results in a higher *Knowledge Consistency* score, ensuring that the extracted causal spans are consistent with annotated causal spans. The *Informativeness* and *Fluency* of our proposed framework is likewise of high quality. These results demonstrate our model's strong ability to understand offensive information and produce results comparable to human annotators.

**Comparison with Existing Works.** Table 4 demonstrates that CMSEKI is the best-performing baseline, which is not unexpected considering that it grasps the input information using common-sense knowledge from external knowledge sources. However, the *DICE* model beats CMSEKI on all measures, especially by 1.95% F1 for the *DI* task and 3 ROS points for the *CE* task. SpanBERT is the highest-performing baseline that does not employ

---

[7]We discuss the definition of each metric in Appendix A.5.2

any external information, outperforming other comparable systems. However, it falls short by 2.88% F1 for the *DI* task and 5 ROS points for the *CE* task when compared to our *DICE* framework. Furthermore, the DICE method managed to outperform the sophisticated state-of-the-art multimodal language models, ViL-BERT CC and Visual BERT COCO. The results analysis reveals that BERT, SpanBERT, and BERT-HateXplain exhibit notably lower performance in the task of cause extraction for offensive content. This observation underscores the inherent difficulty that even powerful language models face when it comes to discerning crucial aspects, such as identifying causes, within offensive content.

| Setup | F1$^{DI}$ (%) |
|---|---|
| $[T+I]_{-T_V}$ | 84.68 (-3.03) |
| $[T+I]_{-V_T}$ | 83.92 (-3.79) |
| $[T+I]_{-DS}$ | 85.74 (-1.97) |
| $[T+I]_{-IMF}$ | 82.76 (-4.95) |
| $[T+I]_{-IMF+DS}$ | 82.11 (-5.60) |
| $[T+I]_{-IMF+DS+AE}$ | 81.40 (-6.31) |
| $[T+I]_{-IMF+DS+AE+VG}$ | 79.98 (-7.93) |
| *DICE (Proposed)* | 87.71 |

Table 5: Results of ablation experiments. The % fall in scores are shown in brackets. IMF: Inter-Modal Fusion, AE: Autoencoder, DS: DeepSentiBank, VG: Vocab Graph, $T_V$: Text infused visual features, $V_T$: Visual infused textual features

**Ablation Study.** To examine the importance of the different modules in *DICE* framework, we remove the constituent components, one at a time, and report the results in Table 5. Specifically, we conduct five ablation experiments: first, we replace the proposed Inter-modal fusion (IMF) mechanism by linear concatenation to fuse multimodal features ($[T+I]_{-IMF}$). Next, we independently eval-

| Model | Text | Label |
|---|---|---|
| **1. Human Annotator** | Colorado school bus driver faces criminal charges for *slapping a 10-year-old student in the face* for not wearing a mask | **Distressed** |
| BERT-HateXplain | Colorado school bus driver faces *criminal* charges for *slapping a 10-year-old* student in the face for not wearing a mask | Distressed |
| SpanBERT | Colorado school bus driver faces criminal *charges for slapping* a 10-year-old student in the face for not wearing a mask | Distressed |
| CMSEKI | Colorado school bus driver faces *criminal charges for slapping a 10-year-old* student in the face for not wearing a mask | Distressed |
| **Proposed** | Colorado school bus driver faces criminal charges for *slapping a 10-year-old student in the face* for not wearing a mask | Distressed |
| **2. Human Annotator** | *If you need #violence to defend against #Jewish ideas, your ideas aren't #terrorism they are #SelfDefence* . Food For Thought. | **Distressed** |
| BERT-HateXplain | If you need *#violence to defend* against #Jewish ideas, your ideas aren't *#terrorism* they are #SelfDefence. Food For Thought. | Distressed |
| SpanBERT | If you need *#violence to defend against #Jewish* ideas, your ideas aren't #terrorism they are #SelfDefence. Food For Thought. | Non-Distressed |
| CMSEKI | If you need *#violence to defend against #Jewish ideas, your ideas aren't #terrorism* they are #SelfDefence. Food For Thought. | Distressed |
| **Proposed** | If you need *#violence to defend against #Jewish ideas, your ideas aren't #terrorism they are #SelfDefence.* Food For Thought. | Distressed |

Table 6: Sample predictions from the various systems

uate the impact of one modality on the other by removing $T_V$ and $V_T$ one by one. We observe from Table 5 that removing the text-infused visual features ($T_V$) has a more detrimental effect on the system's performance compared to removing the visual infused text features ($V_T$). Next, we remove DeepSentiBank sahi kya h(DS) alongside IMF ($[T+I]_{-IMF+DS}$), and, finally, we substitute the proposed IMF, DS and AE mechanism by linear concatenation to fuse multimodal features ($[T+I]_{-IMF+DS+AE}$). We observe a notable fall in scores when either of these modules is removed from the *DICE* approach, especially when we remove the IMF+DS+AE module. This establishes that all components of the *DICE* model developed for multimodal data contribute to the success of the defined tasks in a zero-shot environment.

| Basic Model | $\mathcal{L}_{adv}$ | $\mathcal{L}_{re}$ | $\mathcal{L}_{al}$ | $F1^{DI}(\%)$ |
|:---:|:---:|:---:|:---:|:---:|
| ✓ | | | | 82.98 (-4.73) |
| ✓ | ✓ | | ✓ | 86.01 (-1.72) |
| ✓ | | ✓ | ✓ | 85.51 (-2.20) |
| ✓ | | | ✓ | 83.70 (-4.01) |
| ✓ | ✓ | ✓ | ✓ | **87.71** |

Table 7: Affect of different loss functions. Basic model combines semantic features via zero-shot loss function

To investigate the significance of the loss functions in *DICE*, we remove them one by one and report the results in Table 7. In the first ablated model, we remove all three loss functions (i.e., $\mathcal{L}_{adv}$, $\mathcal{L}_{re}$, and $\mathcal{L}_{al}$). We remove the $\mathcal{L}_{re}$ loss function in the second model and the $\mathcal{L}_{adv}$ adversarial function in the third. In the fourth model, we remove $\mathcal{L}_{adv}$ and $\mathcal{L}_{re}$. When any of these losses is eliminated from *DICE*, we see a performance decline when compared to the proposed method. The performance drop is the largest (4.73%) when all three losses are eliminated. Clearly, loss functions play a crucial role in training the entire model end-to-end.

**Qualitative Analysis.** We thoroughly examined the predictions made by the different systems. Consider the examples in Table 6. The top row displays the tokens (or 'causes') that human annotators noted and that they consider representing the causes. for the post being *Distressed*. The next four rows show the extracted tokens from the various models. We observe that the proposed *DICE* model correctly categorizes the examples as distressed and also extracts good-quality causal spans. In the second example, we observe that although the SpanBERT model extracts a partial causal span correctly, it assigns the wrong label (*Non-distressed*). We also analyze the cases where the proposed model performs poorly. In the interest of space, we present the discussion in the Appendix (section A.6).

## 6 Conclusion

In this work, we present a novel problem of *Distress Identification and Cause Extraction (DICE)* from multimodal online posts. We develop a multitask, deep framework for detecting distress content and identifying associated causal phrases from text using emotional information. We devise a zero-shot strategy to dynamically incorporate emotional information into training and propose a novel fusion mechanism to infuse the features of multimodal inputs. Furthermore, we introduce the first *Distress and Cause annotated Multimodal (DCaM)* corpus, consisting of over 20,764 social media posts. We illustrate the effectiveness of our method by comparing it to several state-of-the-art baselines. When compared to human performance, the present state-of-the-art models perform poorly, which serves to emphasize the difficulty of the task at hand. We believe our work will advance multimodal reasoning and comprehension while also assisting in the resolution of a significant real-world problem.

## Limitations and Future Scope

Due to the low prevalence of hate speech on social media (approximately 3% of messages are hateful), (Fortuna and Nunes, 2018)), we scrape posts by searching for hate words to increase the likelihood of encountering hate-offensive content. This may have invited some undesired sampling bias while constructing the dataset. Additionally, emoticons and other non-standard symbols like $ are often used in current online interactions. One potential research direction is to use these neglected visual features of text information to adapt to more realistic settings.

## Ethical Consideration

We created our resource using publicly accessible social media postings. We adhered to the data use guidelines and did not infringe on any copyright problems. Our Institutional Review Board also reviewed and approved this research. We make the code and data accessible for research purposes through an appropriate data agreement mechanism.

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

# A  Appendix

We discuss the implementation details and present supporting details of the considered baselines and the human evaluation metrics. We also discuss a vivid qualitative analysis that compares our model's predictions with the best-performing baselines.

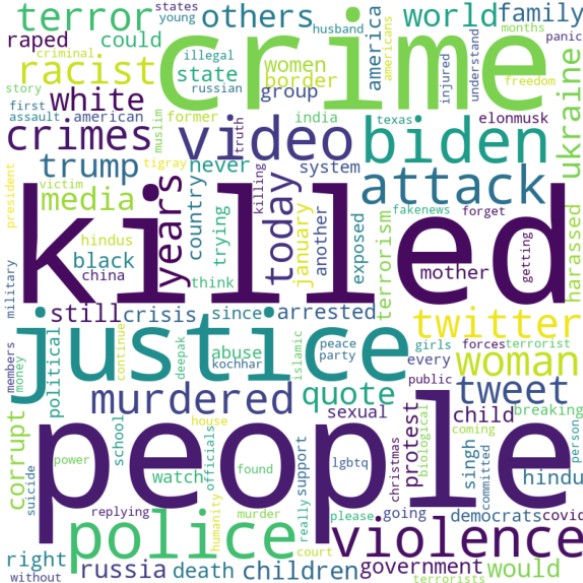

Figure 5: Word Cloud from *Distressed* posts.

## A.1  Word characteristics

We generate word clouds to graphically represent the word frequencies that appear more frequently in the *Distressed* and *Non-distressed* posts. The

bigger the term in the visual, the more often it appeared in user descriptions. Figures 5 and 6 show the word clouds generated from the 100 most frequent words of each class. The difference in word choices for the distinct classes is evident from the figures. Table 8 shows some keywords used for crawling posts from Twitter and Gab to develop the *DCaM* dataset. Initially, we randomly crawled around 5000 posts each for a period of 1 week from both Twitter and Gab and performed topic modeling to fetch the trending topics. We randomly use a subset of these topics to crawl posts for our dataset. From the collected posts, we create a bag of frequently occurring hashtags and use the generated set to crawl further posts. We take care of non-repetition in the collected posts by maintaining the post IDs. Lastly, to supplement the lack of offensive posts being crawled, we use the synonyms of the words 'hate', and 'offensive' and use them as tags (like for the word 'offensive' an example synonym could be 'insult' and gab URL that can be used: `https://gab.com/tags/insult`) to extract posts during data scraping.

| Trending Hashtags |
| --- |
| #MusicBankinChile, #RichaChadha, #SafeFlightOurAstronaut, #Casteist_BCCI, #TheWhiteLotus, #InvestMPinBengaluru, #LopezMartin, #mondaythoughts, #TradersWithDelhiBJP, #RiotGamesONE, #thursdayvibes, #WatchingCricketOnPrime, #NintendoSwitch2022, #AAPProtectsCorruption, #MumbaiAttacks, #MondayMotivation, #KashmirFiles, #IndigoByRM, #pac12championship, #GoBlue, #GujaratElectionResult, #ThePayoff, #TheGameAwards, |

| Trending Topics |
| --- |
| Thanksgiving, Galwan, Shame On You NBT, False News Not Required, Media SoldOut InSSRCase, Fake News Factory, Modiji FastTrack SSRCase, Never Forget, Spiritual Revolution, Cause of Conspiracy, Hurdle For Missionaries, Israel, One Future, Presidency, SSRCulprits Whitewash Shameful, APOLOGIZE TO LISA, Hunter Biden, James Gunn, Christmas, Ukraine |

| Hate-Offensive Synonyms |
| --- |
| toxic, illegal, slave, panic, victim, crisis, assault, protect, terror, protest, teach, justice, sad, alone, broken, feelings, mood, stressed, depression, pain, ugly, attack, shameless, stupid, poverty, revenge, hell, poor, liers, suffer, violence |

Table 8: Sample keywords used for scarping posts to construct *DCaM*.

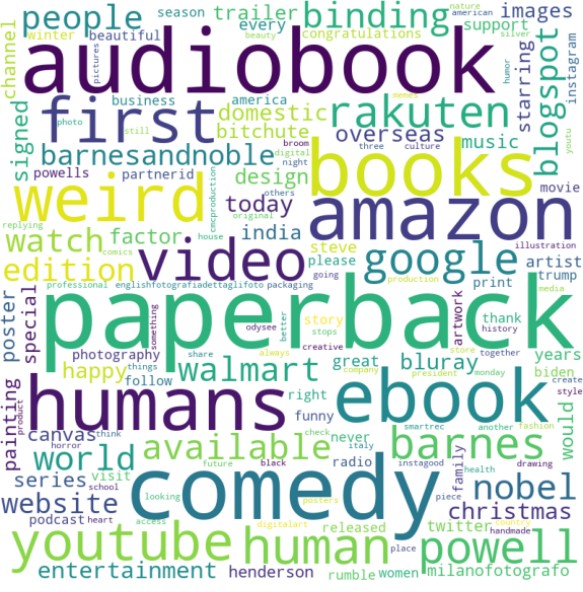

Figure 6: Word Cloud from *Non-distressed* posts.

## A.2 Annotation Guidelines

Our annotation guidelines are rooted in the works of (Poria et al., 2021; Ghosh et al., 2022c). The annotators were instructed to identify the set of causal spans that accurately depict the reasons for a post being tagged as *distressed* given an input post with that label. The annotators annotated a post with the *No_cause* tag if the cause of the post was latent, that is, if there was no stated causal span. Two human experts—graduate students with adequate task knowledge—annotated every post. We used the union of candidate spans from distinct annotators as the final causal span only when the size of their intersection was at least 50% of the size of the smallest candidate span. A third annotator was brought in if the final span could not be determined from the previous spans. This third annotator was similarly told to choose shorter spans over larger spans where they could adequately depict the reason without losing any information.

## A.3 Experimental Setup

We use PyTorch[8], a Python-based deep learning package, to develop our proposed model. We conduct experiments with the BERT import from the huggingface transformers [9] package. To establish the ideal value of the additive angle $x$, which affects performance, five values ranging from 0.1 to 0.5 were examined. The default value for $x$ is 0.30. We set amplification value $a$ as 64. All experiments are carried out on an NVIDIA GeForce RTX 2080 Ti GPU. We conducted a grid search across 200 epochs. We find empirically that our Embedding

---
[8] `https://pytorch.org/`
[9] `https://huggingface.co/docs/transformers/index`

size is 812 bytes. We use Adam (Kingma and Ba, 2015) for optimization. The learning rate is 0.05, and the dropout is 0.5. The auto-latent encoder's dimension is fixed at 812. The discriminator $\mathcal{D}$ consists of two completely linked layers and a ReLU layer and accepts 812-D input features. Stochastic gradient descent has a learning rate of 1e-4 and a weight decay of 1e-3. with a momentum of 0.5. We perform 5 cross-validations of the *DCaM* dataset for training and testing purposes. We run our experiments for 200 epochs and report the averaged scores after 5 runs of the experiments to account for the non-determinism of Tensorflow GPU operations.

## A.4 Baselines

We discuss the details of the considered baselines below. Similar to the *DICE* approach, to adapt the baselines to our multi-task scenario, we add a linear layer on top of the hidden-states output in the output layer of the *CE* task to calculate span start and end logits. The output layer for the *CE* task employs sigmoid activation, in which the threshold value is set at 0.4.

### A.4.1 BiRNN-Attention

The only difference between this model and the BiRNN model is the addition of an attention layer (Liu and Lane, 2016) after the sequential layer. In order to further train the attention layer outputs, we calculate the cross entropy loss between the attention layer output and the ground truth attention.

### A.4.2 CNN-GRU

Zhang et al. (2018) employed CNN-GRU to achieve state-of-the-art on several hate speech datasets. We add convolutional 1D filters of window sizes 2, 3, and 4, with 100 filters per size, to the existing architecture. We employ the GRU layer for the RNN component and max-pool the hidden layer output representation. This hidden layer is routed via a fully connected layer to yield prediction logits.

### A.4.3 BERT

We fine-tune BERT (Liu et al., 2019a) by adding a fully connected layer, with the output corresponding to the CLS token in the input. Next, to add attention supervision, we try to match the attention values corresponding to the CLS token in the final layer to the ground truth attention. This is calculated using a cross-entropy between the attention values and the ground truth attention vector, as detailed in (Mathew et al., 2021).

### A.4.4 ViL-BERT CC

ViL-BERT CC (Lu et al., 2019) is a variant of the ViL-BERT model that has been pre-trained on the Conceptual Captions (CC) dataset. Conceptual Captions is a large-scale dataset containing image-caption pairs sourced from the web. By leveraging the rich and diverse data in CC, ViL-BERT CC is designed to understand and generate captions for images, enabling tasks such as image captioning, visual question answering, and image retrieval.

### A.4.5 Visual BERT COCO

Visual BERT COCO (Li et al., 2019) is a variant of the Visual BERT model that has been pre-trained on the Common Objects in Context (COCO) dataset. COCO is a widely used dataset for object detection, segmentation, and captioning tasks. By pre-training on COCO, Visual BERT COCO learns to encode visual features and understand the context of images, enabling tasks such as object recognition, image captioning, and visual question answering. Visual BERT COCO enhances the model's ability to analyze visual content and perform various vision-related tasks.

### A.4.6 BiRNN-HateXplain and BERT-HateXplain

We fine-tune the models[10] made available by Mathew et al. (2021) on our *DCaM* dataset by changing the output layers as described earlier to suit our task's objective.

### A.4.7 SpanBERT

SpanBERT (Joshi et al., 2020) follows a different pre-training objective compared to traditional BERT system (e.g. predicting masked contiguous spans instead of tokens) and performs better on question-answering tasks. Following the work in (Ghosh et al., 2022c) where SpanBERT is used to solve a mix of classification and cause extraction tasks, we fine-tune the SpanBERT base model on our *DCaM* dataset to meet our objective.

### A.4.8 Cascaded Multitask System with External Knowledge Infusion (CMSEKI)

We contrast the performance of our model with the state-of-the-art CMSEKI system presented in

---

[10]https://github.com/punyajoy/HateXplain

(Ghosh et al., 2022b). CMSEKI leverages common-sense knowledge in the learning process to address multiple tasks simultaneously.

## A.5 Metric Definitions

The following metrics collectively provide a quantitative assessment of how well our model performs in the task of extracting causal spans for manifestations and determinants.

### A.5.1 Evaluation Metrics

- *Full Match (FM)*: This metric measures the percentage of predicted outputs that exactly match the ground truth outputs. In the context of span extraction, it would indicate the proportion of extracted causal spans that are completely correct.

- *Partial Match (PM)*: This metric evaluates the similarity between the predicted outputs and the ground truth outputs, but it allows for some degree of variation. It takes into account cases where only a portion of the prediction matches the ground truth. This can be useful when the extracted causal spans are almost correct but might have minor variations.

- *Hamming Distance (HD)*: Hamming Distance is a measure of the difference between two strings of equal length. It counts the number of positions at which the corresponding symbols in the two strings are different. In the context of causal extraction, it could represent the number of positions where the predicted and ground truth causal relationships differ.

- *Jaccard Similarity (JS)*: Jaccard Similarity is a measure of set similarity that calculates the ratio of the size of the intersection of two sets to the size of their union. In the context of causal extraction, it would assess the similarity between the sets of tokens (or other elements) in the predicted and ground truth sequences.

- *Ratcliff-Obershelp Similarity (ROS)*: The ROS is a sequence comparison metric that measures the similarity between two sequences by identifying the common substrings between them. It calculates a similarity score based on the length of the longest common subsequence between the sequences. This metric would quantify how much of the predicted causal spans match the ground truth causal spans in terms of shared subsequence patterns.

### A.5.2 Human Evaluation-based Metrics

1. *Fluency*: This determines whether or not the extracted span is fluent and natural. Natural and regular answers get a score of 5, whereas inarticulate ones receive a 0.

2. *Knowledge consistency*: This determines whether or not the produced answer has used the appropriate knowledge. If the model generates responses based on irrelevant information, it must get a score of 0, while the selection of pertinent knowledge must receive a score of 5.

3. *Informativeness*: This metric is used to assess how informative the produced replies are. Here, a score of 0 means that the replies are uninformative, and a score of 5 means that they are.

## A.6 Error Analysis

Although our proposed *DICE* framework performs well in the majority of the test cases, still there are certain scenarios where it fails to make the correct predictions. We show some sample predictions from the test set in Table 9. In the first two instances, our model is able to partially predict the causal spans; however, in the first example, it fails to categorize the post as *Distressed*. It is also to be noted that the model extracted span in the second example seems to be more appropriate than the actual annotation by the human annotator. The model rightfully ignores the irrelevant information 'Video shows' and focuses on the relevant action part of the post. This illustrates our model's strong ability to comprehend offensive reasoning among diverse test cases. In the third and fourth examples, our model fails to extract any relevant cause from the given input. Moreover, in the third example, the model wrongly categorizes the post as Non-distressed. This can be due to the lack of sufficient context that hindered our model's comprehension ability for the given input.

| Post | Extracted Span | Predicted Label |
|---|---|---|
| **Partially extracted causal spans** | | |
| **1. Rutgers Professor On White People: " We Gotta Take These MF'ers Out! " #Rutgers #Professor #WhitePeople** | **Take These MF'ers Out!** | **Non-Distressed** |
| **2. EXCLUSIVE: Video shows terrifying ambush-style robbery, shooting in San Francisco #California #Crime** | **terrifying ambush-style robbery, shooting in San** | **Distressed** |
| **Causal spans not extracted** | | |
| **3. Apu reads a children's book about a transsexual Nazi.** | **No_Cause** | **Non-Distressed** |
| **4. Speaking with Raymond Ibrahim about #Christian Persecution** | **No_Cause** | **Distressed** |

Table 9: Error Analysis from the Proposed Systems. Color Coding: Blue- Correct, Red: Incorrect; Teal: Incomplete. Highlighted text in pink shows the human-annotated causal spans.