# OpenReview forum: "Standardizing Distress Analysis: Emotion-Driven Distress Identification and Cause Extraction (DICE) in Multimodal Online Posts"
_EMNLP/2023/Conference — EMNLP 2023 Main_

### Official Review · Reviewer_RS7U · 2023-07-26

**Soundness:** 2

**Excitement:**

3: Ambivalent: It has merits (e.g., it reports state-of-the-art results, the idea is nice), but there are key weaknesses (e.g., it describes incremental work), and it can significantly benefit from another round of revision. However, I won't object to accepting it if my co-reviewers champion it.

**Missing References:**

[1] The Hateful Memes Challenge: Detecting Hate Speech in Multimodal Memes. (NIPS 2020)
[2] SemEval-2022 Task 5: Multimedia Automatic Misogyny Identification. (SemEval 2022)
[3] On Explaining Multimodal Hateful Meme Detection Models. (WWW 2022)
[4] Prompting for Multimodal Hateful Meme Classification. (EMNLP2022)


**Paper Topic And Main Contributions:**

This paper proposes a novel and interesting task of Unified Distress Identification and Cause Extraction. This brings challenges to the understanding and reasoning ability of multimodal models in images and texts.
The paper builds relevant dataset, proposes a multi-task framework, and presents benchmarks of various baselines.


**Questions For The Authors:**

1. What is the proportion of the samples whose cause is latent, i.e. No_cause sample? These samples can be difficult to detect.
2. In Table 4, the F1 and Acc. of DICE are 86.12 and 86.54, which is same as the results of DICE with only textual-modal shown in Table 3. Does Table 4 only show the performance of each model in the textual modal? In the experimental stage, is the input of the ViL-BERT CC and Visual BERT COCO only text? Since these two baselines are multi-modal pre-training models, the introduction of bi-modal as an input will have obvious performance improvement.
3. What is the performance of ViL-BERT CC and Visual BERT COCO in the task of CE?
4. DICE is a general approach for multimodal tasks, so how well does it work on other tasks, e.g. multimodal sarcasm detection?
5. As a multimodal task, the combination of text and image is necessary. However, 199~201 shows that the reason seems to only be based on the text itself, right?


**Reasons To Accept:**

1. Unified Distress Identification and Cause Extraction of multimodal posts is an issue of concern. This paper could lead to a more active discussion of the issue among researchers.
2. The dataset DCaM is on par with other datasets, and presents the cause of some stressed samples.
3. The model DICE, especially Inter-modal Fusion module, is a general approach for multimodal tasks. It has good reusability.


**Reasons To Reject:**

1. Some descriptions in this paper are inconsistent with the actual situation, which may lead to misunderstanding. The authors may have confused hate speech with distress content, believing that the dataset they built contained hate speech (Table 1). However, according to Fig. 2 and samples that have been open-source, most of the distressed samples in this dataset are news reporting violence and discrimination, which is harmless in itself. And it is fundamentally different from hate speech that spreads prejudice and insults others, mostly by user comments.
Therefore, some of the discussions about comparisons with related datasets and methods of hate speech detection may be incorrect. For example, 206-209 mentions the high agreement of the dataset built in this paper and attributes it to the annotation process. In fact, it may simply be that news texts are more objective than user comments, resulting in less difficult and divisive labeling.
In a word, it is suggested that researchers should further clarify their research content, and it is necessary to describe the difference and connection between hate speech and distress content.

2. Illustrations of the model DICE (i.e., Fig. 3 and Fig. 4) are not clear. And many symbols and modules in the figures cannot correspond to the text of methodology. For example,
a. In Fig.3, I can know that Deep SentiBank is introduced in DICE, however, the section of Methodology lacks related introduction.
b. Is the Multi-Modal Fusion module in Fig. 3 be the Inter-Modal Fusion (IMF) in 294 of Section 3.2?
c. In Fig.3, the semantic vectors of emotion classes is represented by BERT, however, 367-369 shows that Word2vec is introduced but not BERT.

3. Existing experiments may not enough to prove the effectiveness of the model DICE:
a. The paper lacks some ablation experiments to prove the effectiveness of the modules, including vocab graph and emotion feature.
b. The paper lacks the necessary baseline experiments, including CLIP, which is the most commonly used multimodal model at present, and BERT+ResNet+MLP, which is the backbone of DICE.

4. The paper lacks some important references about multimodal hateful detection. I will list them later.


**Reproducibility:**

1: Could not reproduce the results here no matter how hard they tried.

**Reviewer Confidence:**

4: Quite sure. I tried to check the important points carefully. It's unlikely, though conceivable, that I missed something that should affect my ratings.

**Typos Grammar Style And Presentation Improvements:**

 Fig. 3 and Fig. 4 are not clear enough and should be replaced by vector diagram.

---

> ### Author Rebuttal · Authors · 2023-08-27
>
> We appreciate the reviewer's thorough review, valuable feedback, and, acknowledge the need for clarity in our paper.
>
> 1. Through our annotation process, we identified that over half of the posts exhibit latent causes, which were consequently labelled as "no_cause." Given this significant portion of the dataset, our proposed DICE framework effectively accommodates data distribution, enabling the model to adeptly address such instances during inference by often predicting empty spans.
>
> 2. We apologize for any confusion caused by the presentation of results in Table 4. Your observation is valid; indeed, ViL-BERT CC and Visual BERT COCO are multimodal baselines, and we trained them on bi-modal inputs. However, their results were mistakenly included in the text-based models section, leading to the confusion. The intention was to present their performance in Table 3, which primarily features multimodal results. In Table 4, we aimed to showcase the results of text-based models, including DICE and other methods, as the remaining baselines do not support multimodal inputs. We will ensure the correction of this inconsistency in the revised version upon acceptance. Thank you for your valuable feedback.
>
> 3. Our DCaM dataset is comprised of image-text inputs, prompting us to incorporate two well-regarded multimodal baselines, ViLBERT CC and Visual BERT COCO, to ensure a comprehensive evaluation. While adapting these models for cause extraction tasks would necessitate modifications to their output layers, we deliberately refrained from making such changes. To maintain fairness in comparison, we ran these models using their original architecture while optimizing the hyperparameters, enabling us to report results specifically for the distress identification task.
>
> 4. While our proposed DICE method is indeed designed to handle various multimodal tasks, our current study's focus was on distress identification and cause extraction from social media posts. We acknowledge the importance of testing the model's robustness on other tasks, including multimodal sarcasm detection, and we are eager to explore such avenues in our future research endeavors.
>
> 5. Our dataset annotations encompass two aspects: the post's distress categorization (Distressed or Non-distressed), where both image and text are considered, and causal span annotation, which predominantly relies on the text. While the image plays a pivotal role in detecting distress, the main objective of the cause extraction task is to extract causal spans from the text, where the role of the image is not central. This distinction underlines the varying roles of text and image in our approach. We trust this clarifies your query.
>
> 6. Our research was driven by a desire to delve into the intricacies of distress classification and cause extraction for distressed posts, with a primary focus on comprehending negative and harmful content efficiently. To streamline the analysis, we grouped various connotations of offensive content under the "distressed" category, as our primary goal was not to categorize among different forms of distress. Our study's central objective was to comprehend the causes associated with various forms of offensive content, including hate, offense, and abuse. Regarding your point about the distinction between hate speech and distressed content in our paper, while Fig 2 might depict a news article post, it's crucial to note that our dataset is deliberately designed to be inclusive of various post sources and types, ensuring a comprehensive representation. Our core objective revolves around identifying any harmful negative content within a post and uncovering its underlying causes. The introduction of the "Distressed" class is a deliberate choice to encompass a wide range of posts that hint at harmful events. This approach aligns perfectly with the overarching aim of our research. We hope this explanation helps clarify our perspective on the matter.
>
> 7. We sincerely apologize for the inconsistencies between the symbols, module names in figures, and the main text. In our revised version, we will rectify these discrepancies to ensure clarity and alignment. Concerning DeepSentiBank (Chen et al., 2014), we acknowledge its significance as a visual sentiment analysis model using deep CNNs. While we omitted a detailed discussion on DeepSentiBank in the original submission due to its established nature, we will incorporate a brief explanation in the revised version, detailing its use for ANP features extraction in our approach. We also acknowledge the confusion regarding the Multi-Modal Fusion module in Fig. 3, which indeed corresponds to Inter-Modal Fusion (IMF) in the main text. To enhance clarity, we will rename it as IMF in the figure. We apologize for the oversight in incorrectly referring to Word2Vec instead of BERT for emotion class embeddings; this will be corrected in the revised version as well.
>
> 8. a) We apologize for not explicitly stating in the main text that the discussion on the ablation experiments (section A.6) are provided in the appendix. We will rectify this oversight in the final version, should it be accepted.
> b) We acknowledge the importance of baseline experiments and will include CLIP and BERT+ResNet+MLP as suggested to enhance the robustness of our study in the revised version.
>
> 9. We will ensure to incorporate the suggested references for a comprehensive literature coverage in the final version if accepted.

---

### Official Review · Reviewer_rcDd · 2023-08-03

**Soundness:** 2

**Excitement:**

2: Mediocre: This paper makes marginal contributions (vs non-contemporaneous work), so I would rather not see it in the conference.

**Paper Topic And Main Contributions:**

Background of the paper:
With the increasing influence of hate speech on social media on public opinion, methods for automatically identifying hate speech have attracted increasing attention. However, past automatic recognition methods have mostly been limited to analyzing textual content, and little attention has been paid to the interpretability of these models, despite the possible social and legal consequences of wrong predictions.

Past solutions:
Past methods mainly focus on the analysis of text content and lack the consideration of multimodal input. In addition, these methods have limitations in terms of interpretability.

Motivation of the paper:
Given the limitations of past approaches, this study poses a novel problem of identifying distress and extracting causes from multimodal online posts. To address this issue, thet develop a multi-task deep framework and use sentiment information to incorporate sentiment information into the training process. Furthermore, they create the first multimodal dataset of distress and cause annotations, demonstrating the effectiveness of their proposed method by comparing with existing benchmarks

**Reasons To Accept:**

- This paper is easy to read.
- They construct a new dataset.

**Reasons To Reject:**

- The method is not novel. 1) Using cross-attention and self-attention to fuse text and image is not new. 2) Using multi-task and incorporating sentiment information to do social media related-task is not new.
- Lack of model ablation. I don't know which technique contributes most to the performance.
- Lack of experiment settings.


**Reproducibility:**

2: Would be hard pressed to reproduce the results. The contribution depends on data that are simply not available outside the author's institution or consortium; not enough details are provided.

**Reviewer Confidence:**

4: Quite sure. I tried to check the important points carefully. It's unlikely, though conceivable, that I missed something that should affect my ratings.

---

> ### Author Rebuttal · Authors · 2023-08-27
>
> 1. We appreciate the reviewer's input. A pivotal aspect of our research involves introducing the novel task of Unified Distress Identification and Cause Extraction (DICE) from multimodal online posts. While cross-attention and self-attention are used in various tasks, our proposed DICE framework introduces a novel inter-modal fusion (IMF) mechanism that dynamically aligns and exchanges information between modalities, facilitating a more comprehensive fusion of text and image features. Our method enhances the adaptability of modal interaction by combining external word embedding models and self-attention mechanisms in the IMF module. Additionally, the incorporation of sentiment information through zero-shot learning allows us to mitigate model complexity and dependency on annotated emotion inputs, setting our approach apart from conventional methods and enhancing its suitability for real-world applications.
>
> 2. We apologize for not explicitly stating in the main text that the experimental setup (section A.3) and ablation experiments (section A.6) details are provided in the appendix. We will rectify this oversight in the final version, should it be accepted.

---

### Official Review · Reviewer_7DRN · 2023-08-05

**Typos Grammar Style And Presentation Improvements:** None.
**Soundness:** 5

**Excitement:**

3: Ambivalent: It has merits (e.g., it reports state-of-the-art results, the idea is nice), but there are key weaknesses (e.g., it describes incremental work), and it can significantly benefit from another round of revision. However, I won't object to accepting it if my co-reviewers champion it.

**Missing References:**

None.

**Paper Topic And Main Contributions:**

introduces an online post-based method for identifying distress and extracting causes.
The primary contributions encompass a multimodal dataset annotated with distress and causes, along with a method for identifying distress and extracting causes.

**Questions For The Authors:**

None.

**Reasons To Accept:**

1. The constructed distress and cause annotated is valuable.
2. The proposed method is well-designed.

**Reasons To Reject:**

1.  Some sections of the paper lack clarity, such as the absence of metric definitions (FM, PM, HD, JF, ROS) on page 7.

**Reproducibility:**

4: Could mostly reproduce the results, but there may be some variation because of sample variance or minor variations in their interpretation of the protocol or method.

**Reviewer Confidence:**

3: Pretty sure, but there's a chance I missed something. Although I have a good feel for this area in general, I did not carefully check the paper's details, e.g., the math, experimental design, or novelty.

---

> ### Author Rebuttal · Authors · 2023-08-27
>
> Aligning with recent research (Ghosh et al., 2022; Singh et al., 2023) in cause extraction, we presented our findings using metrics such as Full match (FM), Partial match (PM), Hamming Distance (HD), Jaccard Similarity (JS), and Ratcliffe-Obershelp Similarity (ROS). We regret the omission of metric definitions and commit to incorporating them in the revised version pending acceptance.

---

### Meta-Review · Area_Chair_aaoR · 2023-09-20

**Recommendation:** 3

**Metareview:**

The paper presents a new multimodal dataset for distress analysis. The social media posts have been annoated for the distress level and the cause. The construction of the dataset is sound and might be a useful resource for the community. The methodology is mostly sound and it might need to clarify some aspects (e.g. metric definitions, detailed experimental settings and model ablation, distress vs hate speech). However, the reviewers did not find the work particularly exciting.

---

### Decision · Program_Chairs · 2023-10-07

**Decision:**

Accept-Main

**Comment:**

The paper presents a new multimodal dataset for distress analysis. The social media posts have been annoated for the distress level and the cause. The construction of the dataset is sound and might be a useful resource for the community. The methodology is mostly sound and it might need to clarify some aspects (e.g. metric definitions, detailed experimental settings and model ablation, distress vs hate speech). However, the reviewers did not find the work particularly exciting.